# On End-to-End Intelligent Automation of 6G Networks

Abdallah Moubayed [1,*] , Abdallah Shami [1] and Anwer Al-Dulaimi [2]

1. Electrical & Computer Engineering Department, Western University, London, ON N6A 5B9, Canada; abdallah.shami@uwo.ca
2. Mobile Solutions Unit, EXFO Inc., Montreal, QC H4S 0A4, Canada; anwer.al-dulaimi@exfo.com
* Correspondence: amoubaye@uwo.ca

**Abstract:** The digital transformation of businesses and services is currently in full force, opening the world to a new set of unique challenges and opportunities. In this context, 6G promises to be the set of technologies, architectures, and paradigms that will promote the digital transformation and enable growth and sustainability by offering the means to interact and control the digital and virtual worlds that are decoupled from their physical location. One of the main challenges facing 6G networks is "end-to-end network automation". This is because such networks have to deal with more complex infrastructure and a diverse set of heterogeneous services and fragmented use cases. Accordingly, this paper aims at envisioning the role of different enabling technologies towards end-to-end intelligent automated 6G networks. To this end, this paper first reviews the literature focusing on the orchestration and automation of next-generation networks by discussing in detail the challenges facing efficient and fully automated 6G networks. This includes automating both the operational and functional elements for 6G networks. Additionally, this paper defines some of the key technologies that will play a vital role in addressing the research gaps and tackling the aforementioned challenges. More specifically, it outlines how advanced data-driven paradigms such as reinforcement learning and federated learning can be incorporated into 6G networks for more dynamic, efficient, effective, and intelligent network automation and orchestration.

**Keywords:** 6G networks; intelligent automation; data-driven opportunities

## 1. Introduction

With the current deployment of 5G networks around the globe, the digital transformation of businesses and services is in full force [1]. This has been exacerbated by the COVID-19 pandemic, which further forced businesses and industries to transition into the digital world at a faster pace [2]. This is illustrated by the projected market size growth of 5G services. More specifically, the global 5G services market size was estimated to be close to 41.5 billion USD in 2020, and is expected to grow at a compound annual growth rate of 46.2%, reaching nearly 665 billion USD by 2028 [3]. Additionally, the 5G market is mainly dominated by the IT and telecom industries (accounting for nearly 25% shares in 2020), with the enhanced mobile broadband applications being the most dominant (estimated at 41% of the market share) [3]. Moreover, the massive machine-type communications (mMTC) applications are expected to have the fastest growth between 2021 and 2028 [3].

This has opened the world to a new set of unique challenges and opportunities driven by the need for communication technologies and architectures that enable growth and sustainability while also promoting the digital transformation at hand. In this context, 6G promises to be the set of technologies and architectures that achieves this goal by offering the means to interact and control the digital and virtual worlds that are decoupled from their physical location. This represents the natural evolution from "people-to-people"-focused communication in 1G–4G technologies through the "people-to-things"-focused communication in 5G, to finally reach the "things-to-things"-focused communication in

6G [4,5]. In turn, this enables the further development of emerging (e.g., intelligent transportation systems (ITSs)) and new (e.g., holographic telepresence) use cases. Accordingly, it is estimated that the global market size for 6G will reach 1773 billion USD by the year 2035 after an anticipated launch date of 2030 [6]. This represents a significant growth in market size of nearly three folds in a space of only seven years (between 2028 and 2035).

Moreover, 6G networks and architectures need to allow for the human experience to be expanded across the physical, biological, and digital worlds. This is due to the emergence of new devices equipped with more intuitive interfaces and new sensing technologies, particularly with the continuous deployment of Internet of Things (IoT) devices [7]. More specifically, these new IoT devices are key components in building smarter cities that aspire to be greener and more sustainable [8]. This is done by deploying these sensing devices for a multitude of applications and use cases, such as water and electricity distribution, healthcare, and transportation [8]. Additionally, these devices can play a pivotal role in automating and improving defense and public safety processes [9]. For example, firefighters can make use of IoT devices installed in buildings to detect fires in a faster and more efficient manner [9]. Similarly, sensors deployed on various platforms such as satellites, radars, and unmanned aerial vehicles (UAVs) can provide situational awareness for military personnel [9]. Moreover, 6G networks are expected to support next-generation industrial operations environments (including Industry 4.0) while considering performance metrics such as positioning, sensing, ultra-reliability, energy efficiency, and extreme real-time [10]. This includes facilitating the predictive maintenance of machines in smart factories, as well as offering automated, customized, and personalized digital services in such environments [11,12].

To this end, 6G will introduce a new set of enabling technologies on top of those currently developed/deployed for 5G networks (e.g., network function virtualization, network softwarization, etc.). This includes novel radio and access technologies such as mmWave and THz-wave communication [13]. Moreover, concepts/paradigms such as cell-less networks, tactile internet, artificial intelligence (AI)-optimized wide-area networks, and the dynamic orchestration of personalized services will be crucial pillars in the successful realization of 6G. As a result, objects ranging from vehicles and industrial machines to wearable devices such as watches and apparel will learn to automatically adapt to our behavior, surrounding environments, and business requirements.

Coupled with the expected benefits (both technological and financial) of developing 6G networks is a fresh set of challenges that need to be addressed and considered. One such challenge is the extremely stringent performance requirements stemming from the combined need for the remote control of devices/machines/entities, realization of augmented reality, and the presence of an immersive media experience [14]. This results in the need for extreme ultra-reliable low-latency (URLLC) performance as well as ultra-high rates of 100 Gbps or higher to facilitate the uncompressed transmission of high-quality 360-degree video. Another challenge is the energy efficiency of any technology, paradigm, or algorithm. This is because the network's performance will be dependent on the available energy in the respective architectural domain (human world, physical world, or digital world). As such, 6G network architectures require:

- Programmable networks;
- Secure, robust, and reliable networks;
- Flexible deployment models;
- User-oriented services and adaptive architecture;
- Efficient and simple automated network orchestration.

Among the many challenges that will arise when considering 6G networks, one emerging challenge is "end-to-end network automation" which includes orchestration as a crucial pillar. The Open Network Foundation defines orchestration as "*the selection of resources to satisfy service demands in an optimal way, where the available resources, the service demands and the optimization criteria are all subject to change*" [15]. In a similar fashion, the European Telecommunications Standards Institute defines orchestration as "*the coordination of the*

*resources and networks needed to set up cloud-based services and applications*" [16]. In the context of 6G, network orchestration brings forward a variety of challenges due to the fact that such networks have to deal with more complex infrastructure and a diverse set of heterogeneous services and use cases. This includes fulfilling both the technical requirements (e.g., performance requirements in terms of throughput and latency, network isolation, etc.) and non-technical requirements (e.g., cost, sustainability, dependability, locality, etc.). As such, orchestration processes need to be more proactive compared to the previously developed reactive methodologies proposed for previous network generations. Moreover, the role of artificial intelligence (AI) and machine learning (ML) technologies as key enablers of predictive/proactive network slicing and orchestration is a paramount issue to consider.

Accordingly, this paper aims at envisioning the role of different enabling technologies towards end-to-end intelligent automated 6G networks. To this end, this paper first reviews the literature focusing on the orchestration and automation of next-generation networks by discussing in detail the challenges facing the efficient and effective management and orchestration of 6G networks. Additionally, this paper presents some of the research opportunities for innovation within this area that aim at addressing the research gaps and tackling the aforementioned challenges. More specifically, it outlines how advanced data-driven paradigms such as reinforcement learning and federated learning can be incorporated into 6G networks for more dynamic, efficient, effective, and intelligent network automation and orchestration.

The remainder of this paper is organized as follows: Section 2 provides an overview of 6G networks and the corresponding enabling technologies. Section 3 summarizes some of the related works addressing 5G orchestration and discusses their limitations. Then, Section 4 delves deeper into one of the main challenges facing 6G networks, namely end-to-end network automation and orchestration. Section 5 presents some of the innovation opportunities worth exploring to tackle the aforementioned orchestration challenge. Finally, Section 6 summarizes and concludes the paper.

## 2. 6G: Preliminaries and Enabling Technologies

### 2.1. Vision and Performance Targets

#### 2.1.1. Vision and Goals

Firstly, 6G represents the next evolution in the digital revolution. It will be the basis for billions of humans, things, and connected triphibian (communicating across land, air, and water) devices that will generate zettabytes of data [17]. Therefore, 6G is expected to be a "self-contained ecosystem of artificial intelligence" that will continually evolve by adopting a human- and machine-centric approach [17]. As such, 6G's aims go beyond just providing faster mobile internet access. Rather, its goals are [17]:

1. **Converging the digital, physical, and human world** by supporting digital twinning (tight synchronization between domains to achieve digital twins), immersive communication (extending human senses by providing high-resolution visual/spatial, tactile/haptic, and other sensory data to offer an immersive experience), cognition (being aware of humans' intentions, desires, and moods), and connected intelligence (providing trusted AI functions that can act in and on the network).
2. **Providing network flexibility and programmability**. This includes being able to efficiently internetwork between IoT devices and the communication network, support distributed edge computing solutions, and design protocols suitable for cloud support.
3. **Supporting deterministic end-to-end services**. This is particularly important given the need for 6G networks to support new services and applications such as remote-control and tactile internet.
4. **Emphasizing sustainability** by assisting in reducing energy and emissions' footprint. This can be done by suppressing the energy consumption increase while maintaining the data traffic increase.

5.  **Providing integrated sensing and communication** (for high-accuracy localization and high-resolution sensing services). This will allow for new ways to harvest and interpret the "communication context", particularly for applications and services such as e-health, autonomous vehicles/drones, and advanced cross reality.

6.  **Providing trustworthy infrastructure** (the basis of future societies). This includes keeping terminals' locations private and ensuring that all the network entities (network functions, operating systems, hardware platforms, etc.) are continuously formally verified. As a result, this will build a trusted 6G network.

7.  **Being scalable and affordable** to ensure global coverage across the world. This can be done by ensuring that the system is global (utilized worldwide), the number of interfaces is reduced, and the cost of mobile devices (with basic functionalities) can be lowered.

Figure 1 summarizes 6G's vision and its corresponding goals.

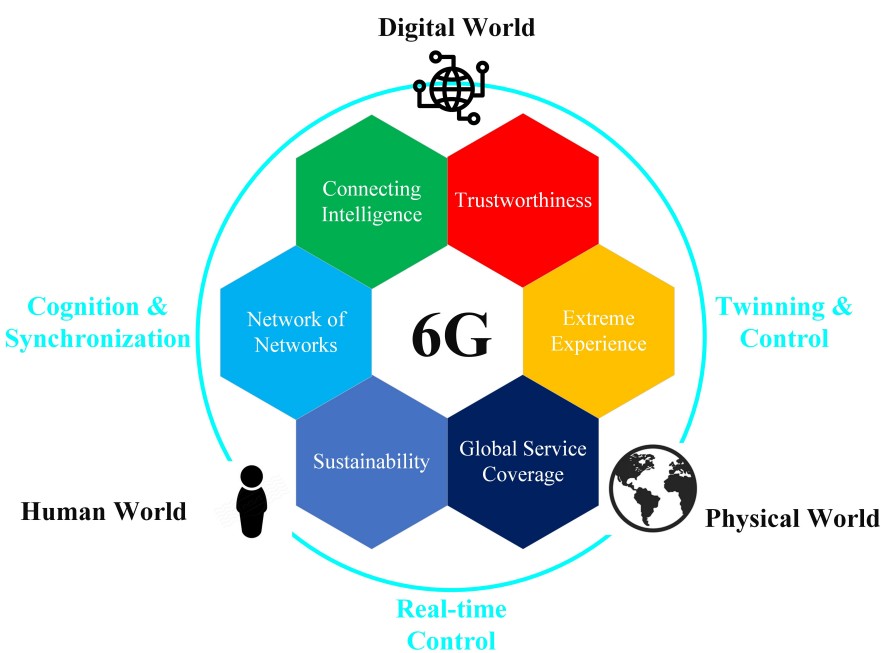

**Figure 1.** The 6G vision.

2.1.2. Performance Targets and Resulting Paradigms

Given that 6G is expected to support new applications and services (such as holographic telepresence, digital twins, telemedicine, etc.) in addition to those offered by 5G, more stringent performance targets are expected [18]. These requirements will be more diverse compared to 5G, with some applications and services requiring a combination of them. In what follows, a brief overview of these performance requirements and some of their applications/services/use cases is given and is summarized in Figure 2.

- **Time-Sensitive Communications:** End-to-end low latency is one of the major performance requirements for 6G to meet. More specifically, 6G is expected to achieve a latency of around 1 ms or lower, particularly for applications such as autonomous/connected vehicles within ITSs, autonomous customer service, and manufacturing. This is done with the goal of offering more advanced real-time interactive AI-enabled services [19].

- **Extremely High Data Rate/Capacity:** The second performance requirement that 6G networks strive to achieve is extremely high data rates and high-capacity communication. More specifically, 6G is expected to achieve peak data rates exceeding 100 Gbps with $100\times$ more capacity. In turn, this would allow for new sensory services that mimic and even surpass the humans' five senses, achieving what is known as "multisensory communication". With this, users will be able to share their virtual experiences and

collaborate virtually in the cyberspace [20]. Additionally, industrial use cases will also benefit from this, particularly in the uplink direction when a large amount of data is sent to the AI-enabled cloud.

- **Extremely High Reliability:** Another performance requirement that 6G is expected to meet is extremely high reliability. More specifically, 6G is expected to offer highly secure, private, and resilient networks that can offer guaranteed quality of service (QoS) up to 99.99999%. This requirement is particularly important for multiple services and use cases [21]. For example, having such a reliable communication network can help to remotely control and automate factory functioning in real time. In addition to the reliability aspect, 6G networks are expected to have high security and privacy. This is because cyber-attacks such as spoofing, falsification, denial, and unauthorized operations can lead to theft, property/personal information leakage, and service suspension. Moreover, such attacks can result in accidents, threatening the lives of people, devices, and systems.

- **3D Dimension Coverage:** The fourth requirement for 6G networks is 3D dimension coverage, also commonly referred to as extreme coverage. This means that 6G networks are expected to expand their coverage area in all environments (including the sky, sea, and space) while still maintaining Gbps-level communication speeds. The idea is to offer communication service coverage in environments with no current human existence. As a result, new services and industries can be supported, such as drone-enabled home delivery as well as primary industries such as forestry and fisheries. Moreover, futuristic applications (e.g., space travel) can be supported with this requirement [22].

- **Extreme Massive Connectivity:** Another crucial requirement for 6G networks is extremely high connectivity. This is due to the substantial growth in the use and deployment of wearable, sensing, and other IoT devices in the real world. As such, 6G networks are expected to be able to support up to 10 million devices/km$^2$ [23]. This represents a 10-fold increase in the number of connections compared to 5G. In addition to the larger number of connections, 6G is also expected to provide added sensing and high-precision positioning (cm-order) capabilities. This is expected to be achieved by fusing AI and wireless communication technologies to help identify objects as well as offer highly precise object detection.

- **Extremely Low Energy and Cost:** Lastly, extremely low power consumption and cost reduction is a crucial requirement for 6G networks and devices [24]. This is because 6G networks are expected to be a pillar in creating sustainable cities and societies. This is important both from the business (reduced capital and operational expenditures) and environmental (highly green communication) point of view. As such, it is envisioned that 6G devices may not need batteries themselves. Rather, 6G devices may harvest the energy they need from the wireless signals within their environment using the combination of advanced communication technologies and novel AI modules with the goal of having zero-energy devices [25].

As a result of the aforementioned performance targets, new concepts/paradigms will emerge. This includes concepts/paradigms such as cell-less networks, AI self-automated networks, and tactile internet. These concepts/paradigms are dependent on the achievement and surpassing of the performance characteristic targets illustrated in Figure 2.

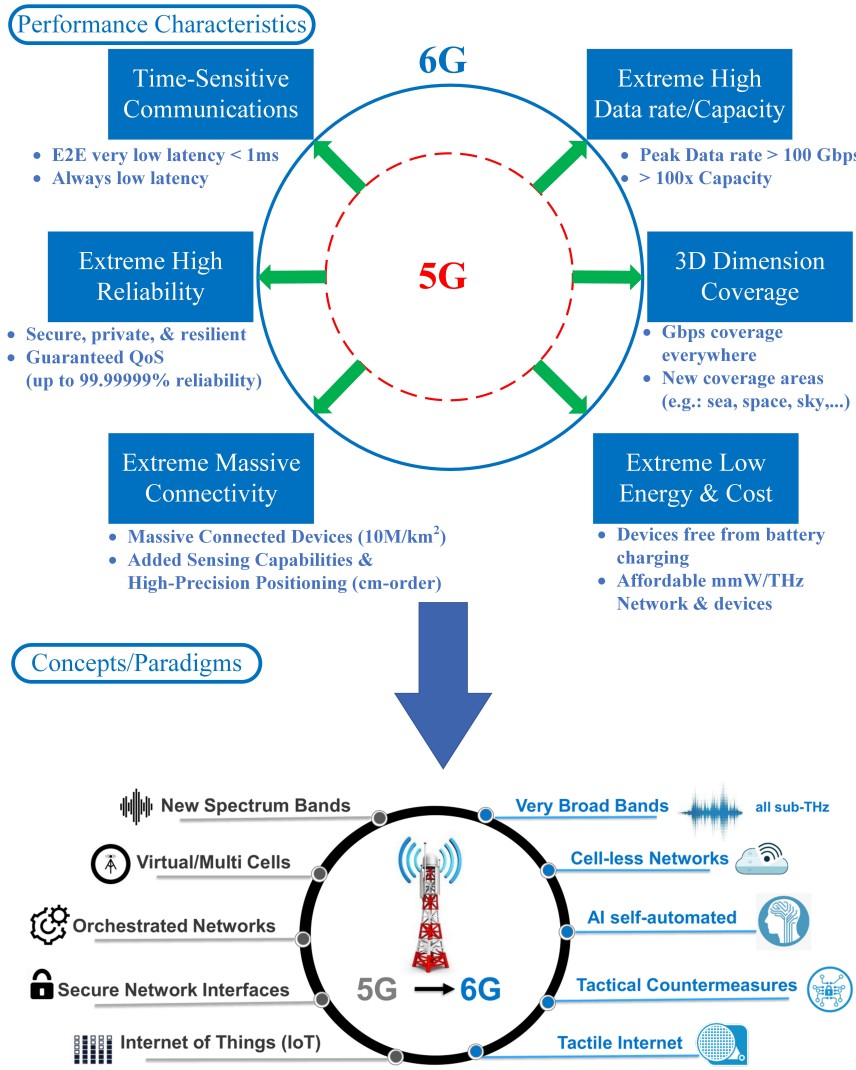

**Figure 2.** The 6G evolution requirements compared to 5G in terms of KPIs and concepts/paradigms.

*2.2. Enabling Technologies*

Furthermore, 6G networks will rely on a multitude of new and innovative technologies to support its vision and achieve its goals. These technologies will be adopted and deployed at different levels of 6G networks, ranging from the antennas and wireless communication technologies all the way to the network architecture and integrated technologies. Figure 3 summarizes some of the key enabling technologies of 6G networks.

2.2.1. Wireless Communication Technologies

One essential aspect of 6G networks is the underlying wireless communication technologies to be used. These technologies are the cornerstone that enables the achievement of the target performance requirements of 6G networks. To complement the currently deployed wireless communication technologies, new technologies such as millimeter-wave (mmWave), terahertz (THz), and optical wireless communication (OWC)/visible light communication (VLC) are being proposed as supporting technologies in 6G networks.

- Millimeter-Wave (mmWave) Communication: mmWave communication is defined as communication using the frequency band ranging between 30 GHz and 300 GHz [26]. Due to the frequency band used, mmWave communication has the ability to offer Gbps speeds [27]. Accordingly, it has already been proposed as one of the technologies for 5G networks, particularly using frequencies below 50 GHz [28]. For example,

mmWave has been proposed for potential deployment for enhanced local area access in 5G, achieving a peak data rate exceeding 10 Gbps [29]. Similarly, it has been proposed for cellular access for distances up to 200 m, achieving a 20× increase in capacity compared to 4G when utilizing directed antennas [29]. However, to support 6G and meet its performance requirements, further bands (such as above 100 GHz) will be needed to achieve the target throughput and latency values. More specifically, mmWave is expected to play a crucial role for short-distance communication scenarios such as in autonomous/connected vehicles and smart factories communication [30]. It is worth noting that adopting mmWave introduces some challenges pertaining to the potential channel blocking by nearby entities (e.g., nearby vehicles in the case of autonomous/connected vehicles), which results in lower channel reliability.

- Terahertz (THz) Communication: THz communication is defined as communication using the frequency band from 0.1 to 10 THz [31]. Compared with mmWave, the benefit of using THz communication is two-fold. Firstly, there is an abundance of available spectrum in this frequency band (larger than that of mmWave communication), thus allowing for more users. Secondly, THz communication has the ability to support even higher throughput rates, reaching Tbps [32]. This can be done without the need for any additional techniques for spectral efficiency enhancement [33]. Moreover, it offers improved system reliability, better energy efficiency, and higher robustness/adaptability to changing propagation scenarios [33]. Accordingly, THz communication is being proposed for multiple throughput-demanding delay-sensitive applications such as sensing, imaging, and localization [33]. Additionally, in a similar fashion to mmWave communication, THz communication is being proposed for a multitude of applications and use scenarios, such as in local area networks, military communication applications, wireless data center networks, and space communication networks [34]. Despite the many benefits offered by THz communications, many corresponding challenges also arise. For example, one major challenge is the transceiver design and the high associated cost, particularly for the digital-to-analog and analog-to-digital converters (DACs and ADCs) [35]. Another challenge is the channel and noise modeling at the THz spectrum. This is because the channel characteristics in this frequency band significantly differ from other bands [35]. Similarly, due to the wavelength size, THz signals suffer from noise resulting from molecular absorption [35]. Hence, these concepts need to be considered when using THz communication for 6G.

- Optical Wireless Communication (OWC)/Visible Light Communication (VLC): As the name suggests, optical wireless communication (OWC) is defined as communication using the optical frequency band, namely between 300 GHz and 30 PHz [36]. This represents a rich source of bandwidth availability equivalent to 400 THz, including the infrared, visible light, and ultraviolet sub-bands [36]. Among the potential OWC systems to be adopted, visible light communication (VLC) is considered to be an important supplementary technology to the current radio frequency communication systems. VLC is communication using the frequency band between 400 THz and 800 THz [37]. Similar to the THz communication, the benefit of using VLC technology as a supplementary technology is two-fold. First, it offers high data transmission rates (up to 10 Gbps per wavelength [38]). Second, it can offer illumination solutions for indoor environments, particularly given the abundance of off-the-shelf optical devices available [38]. An added benefit of VLC that THz communication cannot offer is added security. More specifically, VLC can offer high physical layer security [39]. This is in contrast to THz communication, which can only offer partial physical layer data security [40]. Due to its characteristics, VLC is being proposed as a viable solution in multiple applications, such as indoor LiFi (light-based WiFi), vehicle-to-vehicle communication, and underwater communication, to name a few [37]. Accordingly, OWC in general and VLC specifically promise to play a major role in 6G networks to enable and facilitate the achievement of the desired goals. This should be done by taking into account some of the limitations caused by the deployment of OWC and

VLC-based systems, such as frequent handovers, high inter-cell interference in dense deployments, high potential flickering, and the unsuitability for outdoor environments due to atmosphere-induced losses [41].

- Open Radio Access Network (O-RAN): Another crucial technology that will significantly contribute to 6G is O-RAN. O-RAN is a deployment approach for mobile fronthaul and midhaul networks that is completely disaggregated and reliant on the principles of cloud native applications [42,43]. It is *"a concept based on interoperability and standardization of RAN elements including a unified interconnection standard for white-box hardware and open source software elements from different vendors"* [43]. As a result, the base station is transformed into a modular software stack that can run on commercial off-the-shelf hardware [43]. In turn, this allows different suppliers/companies to provide baseband and radio unit components that can work seamlessly together [43]. Accordingly, the main benefit of adopting O-RAN architecture is that it has the capability of unleashing novel levels of innovation by facilitating and reducing the RAN market entry requirements to new competitors [44]. Consequently, the O-RAN architecture is being proposed heavily as part of the development efforts of 6G networks [45].

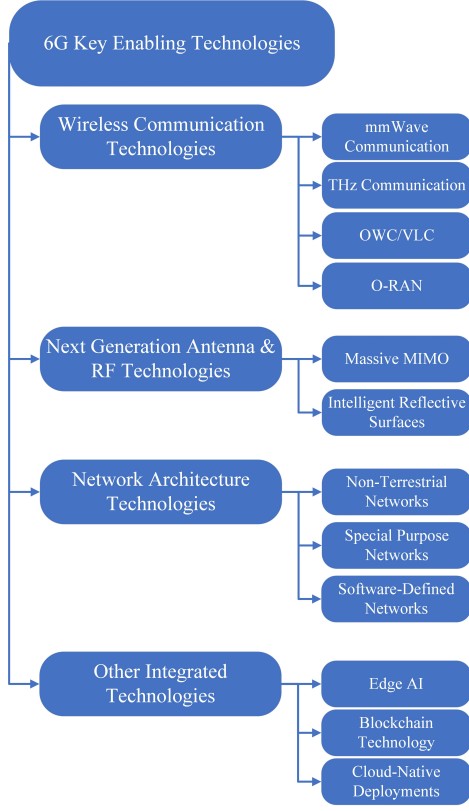

**Figure 3.** The key 6G enabling technologies.

### 2.2.2. Next-Generation Antenna and RF Technologies

Another component that will be vital in 6G networks is the underlying physical device architectures and corresponding environments. These play a crucial role in the actual transmission and reception of the wirelessly communicated data. As such, technologies such as massive MIMO and intelligent reflective surfaces will be key enablers of 6G networks.

- Massive MIMO: Massive Multiple Input Multiple Output (mMIMO) is a concept first proposed by Marzetta as part of the efforts to meet the exponential growth of traffic demand [46]. The concept refers to the use of "physically small, individually controlled antennas to perform multiplexing and demultiplexing for all active users by utilizing directly measured channel characteristics" [46]. Thus, it makes use of spatial multiplexing to allow the various users to occupy the same frequency and time slots

without interference. Based on this fact, mMIMO has already been proposed as part of the development of 5G networks [47]. However, to meet the requirements of 6G, the concept of ultra mMIMO is emerging, in which the large arrangement of antennas is deployed over a larger surface area (e.g., the side of a building). This allows for higher levels of connectivity and throughput [48].

- Intelligent Reflective Surfaces: As a promising paradigm for 6G to achieve a reconfigurable wireless propagation environment, an intelligent reflective surface (IRSs) is a planar surface composed of a large number of passive reflecting elements [49]. Each element of this surface can induce a controllable amplitude and/or phase change to the incident signal independently [49]. As a result, the propagation channel between the transmitter and receiver can be flexibly reconfigured, allowing for potentially lower interference and better reliability [49]. Due to its characteristics, IRS is being proposed as a key solution for various challenges and applications. For example, IRS can be used to communicate with users located in dead zones or at cell edges by creating virtual line-of-sight (LoS) links between the users and the corresponding base stations [50]. Similarly, IRSs can enable the massive D2D communication while simultaneously mitigating the corresponding interference [50]. Lastly, IRSs can facilitate simultaneous wireless information and power transfer (SWIPT) to multiple IoT devices [50]. Accordingly, IRSs are expected to play a pivotal role in offering extreme coverage and extreme massive connectivity in 6G networks.

### 2.2.3. Network Architecture Technologies

The third component introduced by 6G networks that has a direct impact on the orchestration process is the new network architecture topologies and technologies. This because these new network architectures need to be managed by the orchestration module. As such, their characteristics need to be considered when orchestrating 6G networks. Among these new network topologies and technologies are non-terrestrial networks and special purpose networks.

- Non-Terrestrial Networks: As the name suggests, non-terrestrial networks are defined as the set of networks that are deployed off-land. More specifically, it is the set of networks deployed under water, in the air, or even in space [51]. One example of such networks is unmanned aerial vehicle (UAV) networks [52]. In these networks, these UAVs or drones can either act as base stations, cellular users, or signal relays [52]. The main role of these NTNs is providing extended coverage to areas that are generally inaccessible or have harsh conditions. As such, by combining these NTNs with terrestrial networks, the vision of extreme coverage and extreme connectivity in 6G can be achieved.
- Special Purpose Networks: Another important type of networks that is emerging as a result of 6G is special purpose networks, commonly referred to as "sub-networks" [53]. These networks typically belong to one vertical industry, such as industrial automation, autonomous vehicles, and health-focused body area networks [54]. The role of such networks is to achieve the goal of converging the digital world with the physical and human worlds. As such, they are key enablers of 6G networks given that they offer added connectivity and potential intelligence to 6G networks.
- Software-Defined Networks (SDN): In addition to the aforementioned newly emerging networks, another enabling technology that already exists and will continue to play a pivotal role is Software-Defined Networks (SDNs). The basic premise of SDNs is separating the control and data planes using a centralized logical intelligence [55]. As a result, network management is facilitated, and programmability is introduced, in addition to flexibility, scalability, and robustness being offered to the network [55].

Within the context of 6G, SDNs continue to be adopted and proposed as a key network architecture choice [56]. More specifically, the SDN controller is being proposed as the "brain" within the control layer of 6G networks [56]. This component is being extended to the non-terrestrial portion of 6G networks, such as space, air, and even sea-based networks.

Hence, SDNs are still considered to be crucial to 6G networks by extending the paradigm to the new network types and components. However, a key challenge/issue to consider, particularly in 6G networks and the corresponding diverse sub-networks comprising them, is the choice of SDN deployment architecture. More specifically, choosing between a centralized controller architecture (better orchestration/management decisions at the expense of higher signaling overhead and less reliability) and a distributed controller architecture (less signaling overhead and better reliability at the expense of less optimal orchestration decisions) is a critical choice as it can have a direct impact on many of the desired performance metrics [57].

### 2.2.4. Other Integrated Technologies

In addition to the aforementioned technologies, other novel technologies will be integrated into 6G networks to help successfully realize the vision and goals of such networks. This includes technologies such as edge AI and blockchain technologies. These technologies will be pivotal in ensuring that 6G networks are intelligent and secure.

- Edge AI: With the plethora of data and computing resources becoming available, the AI and ML paradigms have gained traction in recent years [58]. In particular, deep learning methods are being increasingly adopted and deployed for various applications, ranging from speech recognition to computer vision, among others. Additionally, there is increased interest in integrating AI and ML (such as deep learning techniques) into the wireless communication field, particularly to help manage and allocate the wireless resources in 6G networks [59]. This is mainly due to the network heterogeneity of the networks and the extremely large data volume expected in 6G. Thus, to achieve autonomous network management and control, AI techniques need to be deployed both in a centralized and distributed manner to enable both local and collective network intelligence. As such, AI deployed at the edge (commonly referred to as "edge AI") using paradigms such as deep learning and federated learning will be a key enabler in 6G networks by autonomously sustaining high KPIs and managing the network's resources and functions.

- Blockchain technology: Another promising technology that has been garnering interest and being increasingly deployed is blockchain technology. Blockchain is one example of what is known as "Distributed Ledger Technology" (DLT) [60]. The concept was first introduced by Stuart Haber and W. Scott Stornetta in 1991 as a mechanism to provide a time-stamp to a digital document [60]. In this mechanism, blocks (which can be thought of as a list of data structures) are connected and securing using cryptography [60]. As a result, this guarantees the distribution of the information in a decentralized manner to avoid any potential tampering. Hence, blockchain promises to provide decentralization, immutability, and transparency [60]. Due to its characteristics, blockchain is being proposed as a key enabler for 6G networks to ensure their security and privacy [61]. More specifically, blockchain can offer intelligent resource management, elevated security features (such as privacy, authentication and access control, integrity, availability, and accountability), and scalability [61].

- Cloud-native deployments: In addition to the aforementioned integrated technologies, cloud-native deployments are essential pillars for 6G networks. As per VMware's definition, *"Cloud native is an approach to building and running applications that exploits the advantages of the cloud computing delivery model [62]"*. This spans a multitude of technologies and tools, including network function virtualization (NFV). NFV will continue to play a pivotal role in 6G networks [63,64]. Simply put, NFV can be defined as the migration process from dedicated hardware to software-based applications/containers/dockers that can run on standard commercial-of-the-shelf servers [65]. The benefit is that it improves network flexibility and scalability, reduces development cycles, lowers capital and operational expenditures (CAPEX and OPEX), and makes platforms more open [65].

Due to the characteristics of the NFV paradigm and the corresponding benefits that it offers, it has been proposed as a pillar and key enabler of a multitude of applications and services to be offered by 6G networks [66–68]. For example, various virtual network functions (VNFs) are being deployed as part of 6G network monitoring, resource management, and security modules of terrestrial and non-terrestrial 6G networks [66–68]. As such, NFV is and will continue to be an essential paradigm as part of any future 6G network deployment. This should be done while taking into account the different challenges introduced by adopting the NFV paradigm in 6G networks. This includes the functional and service-level considerations (placement and orchestration decisions), heterogeneity, and security considerations (redundancy and reliability) of virtual network functions [69].

## 3. 5G Orchestration: Related Works and Limitations

To better understand the challenges facing orchestration in the context of 6G networks, it is important first to explore current state-of-the-art orchestration frameworks and solutions proposed and deployed in 5G networks. Additionally, it is crucial to discuss their limitations/shortcomings as this will form the basis for 6G orchestration frameworks to try to solve.

As part of the related work exploration process, the Scopus database was used to ensure that non-refereed publications are excluded. As a result, the papers chosen are mainly from top-tier conferences and journals of reputable publishers, such as IEEE, Springer, Elsevier, and MDPI. The search terms used are "5G" AND "Orchestration", with the search period limited to the years between 2018 and 2022 to represent the state-of-the-art (SOTA) works.

### 3.1. Related Works

Sgambelluri et al. focused on the problem of network service provisioning and orchestration in 5G networks [70]. More specifically, the authors proposed a multi-domain orchestrator (MdO) prototype based on the 5G exchange (5GEx) project [70]. The goal is to be able to orchestrate services across multi-domain and multi-technology 5G networks. To that end, the authors built on the functionalities offered by SDN and NFV paradigms to design and implement the 5GEx-MdO prototype that is capable of [70]:

- Discovering other operators' orchestrator and related capabilities/resources;
- Receiving and deploying network services and resource requests from other orchestrators;
- Monitoring the deployed requests/services' performance.

Experimental results showed that the proposed prototype is able to deploy new orchestrators in less than two minutes and instantiate new network services and requests in less than 2 s [70].

Similarly, Rostami et al. also built on the functionalities of SDN and NFV as part of their 5G network orchestration solution [71]. However, the authors focused on the orchestration of the transport network and radio access network (RAN) [72]. To that end, the authors proposed a hierarchical cross-domain orchestrator that offers network programmability and flexibility. To make the required decisions, the orchestrator monitors the radio resources at the network access edge level, the transport resources at the access and aggregation levels across multiple domains (e.g., optical, packet, and microwave networks), and the cloud resources at the network core level. Experimental results using their testbed showed that the proposed orchestrator improved the service agility and efficiency of resource utilization, particularly in the case of multiple service providers [71].

Baranda et al. also addressed the problem of multi-domain multi-technology 5G network orchestration [73]. More specifically, the authors focused on orchestrating 5G crosshaul (integrated fronthaul and backhaul) transport resources with the goal of providing automated, integrated, and flexible 5G network management. To that end, a hierarchical approach that exploited a developed application programming interface (API) was adopted to automate network resource orchestration. Accordingly, the authors deployed a complete 5G crosshaul network instance on a geographically distributed testbed. Experimental

results showed that the proposed orchestrator was able to set up a complete service in around 10 s. Additionally, it reduced the local service recovery time to around 0.3 s and the centralized service recovery time to around 6.6 s. Hence, the proposed orchestration solution met the 5G network service setup time target of minutes [73].

Li et al. also adopted an SDN-based approach for 5G network orchestration [74]. The authors focused on integrating edge-centric computing (ECC) and content-centric networking (CCN) paradigms, two promising technologies in 5G networks, into a hierarchical structure titled ECCN [74]. To that end, the authors proposed a heterogeneous RAN architecture that supports both ECC and CCN paradigms. Additionally, the authors designed a specific SDN-based protocol that decoupled the data forwarding and network control of ECC and CCN. Simulation results showed that the proposed ECCN orchestration scheme improved the hit ratio of CCN networks and reduced the amount of redundant traffic sent from the cloud core to the edge devices. As such, more network resources are available for users to request new services and content [74].

On the other hand, Antevski et al. focused on the problem of 5G transport network resource orchestration for vertical industries [75]. To that end, the authors proposed an orchestrator architecture that supports the underlying vertical services with the goal of meeting their diverse resource and service requirements. Accordingly, the authors formulated the problem of resource orchestration as a binary integer programming problem with the objective of minimizing the cost and delay of the vertical services to be hosted while meeting the capacity and delay requirements [75]. Additionally, the authors used three low-complexity algorithms, the first being a graph clustering-based algorithm, the second being a greedy distance-based algorithm, and the third being a greedy delay-based algorithm. Simulation results showed that the proposed strategies were able to determine suitable solutions that provided a good trade-off between the orchestration cost and service delay.

Hoang et al., however, focused on the optimal cross-slice orchestration problem [76]. More specifically, the authors formulated the problem of network slice orchestration based on the service requirements and available resources as a Markov Decision Process (MDP). Accordingly, the authors aimed at finding the optimal policy for 5G networks' cross-slice admission control and resource allocation. Simulation results showed that the proposed framework successfully provided a slice-as-a-service solution that satisfied the service requirements and simultaneously maximized the provider's revenue.

Dieye et al. also focused on the problem of multi-domain network provisioning [77]. Within this problem, the authors considered automated resource sharing and cooperative resource provisioning. To that end, they proposed an auction-based NFV orchestrator. Within this framework, inter-operator interactions and network resources' exchange are considered to be buyer/seller transactions [77]. This was motivated by the economic efficiency of such an approach, particularly given that it facilitated the automatic discovery of service chain market value and assignment of limited resources to the bidders who value them the most. Accordingly, the authors developed a distributed multi-agent deep reinforcement learning framework in which the agent learns strategic actions that maximize the service providers' (SPs) profits. Simulation results showed that the proposed framework outperformed other learning-based and learning-free methods in terms of SPs' profits [77].

Wang et al. focused on the orchestration of the 5G SDN-based network slicing problem [78]. To that end, the authors proposed a hybrid metaheuristic-based solution that combined genetic algorithm (GA) and particle swarm optimization (PSO). In the proposed solution, the GA portion of the algorithm is used to update and optimize the network slices while the PSO portion is used to find the local/global optimal solution and determine the optimal network slice [78]. Accordingly, the authors chose the delay and bandwidth as key performance indicators of 5G networks and used them as part of the fitness evaluation function of the proposed hybrid GA-PSO algorithm. Experimental setup results showed that the proposed hybrid orchestration algorithm achieved good performance, particularly in high-load traffic scenarios of large-scale networks.

In contrast, Singh et al. focused on 5G RAN energy efficiency when developing their

orchestration solution [79]. This was done since energy consumption is a major concern for 5G telecom operators, particularly given the introduction of paradigms such as RAN disaggregation, virtualization, and cloudification. To that end, the authors first formulated the problem of virtual RAN (vRAN) energy consumption optimization as an integer quadratic programming problem [79]. Additionally, the authors developed an innovative and computationally efficient energy-aware orchestrator that leveraged Lagrangian decomposition and simulated annealing algorithms. Using a real-world mobile network traffic dataset for a large metropolitan city, the authors' performance evaluation showed that the proposed energy-efficient orchestration framework achieved energy efficiency gains of up to 25% and 42% compared to state-of-the-art and traditional distributed RAN configurations, respectively. This highlighted the proposed framework's effectiveness, efficiency, and suitability for 5G metro-scale RAN networks [79].

Thiruvasagam et al. again focused on the optimal network slicing problem as part of the 5G network orchestration process [80]. However, the authors in this case focused on the resiliency and latency awareness of the orchestration decisions. To that end, the authors formulated the problem of resilient and latency-aware 5G network slice orchestration in a mobile edge computing (MEC)-enabled environment as a binary integer programming (BIP) problem. The objective of the BIP model was to reduce the deployment cost of the different slices while taking into consideration the bandwidth, latency, and placement requirements/constraints. Moreover, since the BIP problem is shown to be NP-hard, the authors proposed a GA-based solution for the aforementioned problem [80]. Simulation results showed that the proposed GA-based solution reduced resource wastage, improved network throughput, and provided high resiliency against failures [80].

Note that these works can be categorized using one of two points of view: the first being the methodology adopted to solve the orchestration problem and the second being the level at which the orchestration decisions are made. Table 1 summarizes the related works in terms of the methodology adopted, including "Mathematical Optimization-Based", "Game-Based", "Metaheuristic-Based", or "Heuristic-Based" solutions. In a similar manner, Table 2 summarizes the works in terms of the orchestration decisions' level being at the "Core", "Transport", or "Radio Access" network level.

**Table 1.** Summary of previous efforts—5G orchestration methodology.

| 5G Orchestration Technique | List of Previous Efforts |
| --- | --- |
| Mathematical Optimization-Based Solutions | [75,77,79,80] |
| Game-Based Solutions | [76,77] |
| Metaheuristic-Based Solutions | [78–80] |
| Heuristic-Based Solutions | [70,71,73–75] |

**Table 2.** Summary of previous efforts—5G orchestration level.

| 5G Orchestration Level | List of Previous Efforts |
| --- | --- |
| Core Level | [70,76–78,80] |
| Transport Network Level | [71,73,75] |
| Radio Access Network Level | [71,74,79] |

### 3.2. Limitations

Many of the previous works on 5G network orchestration that have been previously proposed in the literature and developed by the industry suffer from multiple limitations/shortcomings. In what follows, a brief discussion of these limitations/shortcomings is presented.

1.  **Centralized Orchestration:** One main shortcoming in current 5G network orchestration frameworks is their centralized nature. Most current frameworks do not extend

to the extreme edge. More specifically, the current orchestration solutions rely on ETSI's NFV MANO framework [81,82]. This framework is typically deployed in a centralized location. This is a major limitation, particularly as it may represent a single point-of-failure and cause significant congestion at the core network level. Hence, using it as the basis of current 5G orchestration solutions is a significant concern.

2. **Single-Layer Orchestration**: Another limitation facing 5G orchestration frameworks is that they often only focus on orchestrating at one particular level. For example, the focus can be only on the radio level, transport level, or core level [83]. This is highlighted by the related works summarized above, in which the focus was often on orchestrating one type/level of the network. The problem lies in the fact that orchestrating at one level only may not result in the optimal allocation and utilization of all the available resources, particularly given that the resources at the different levels of the network are dependent and have a direct impact on the overall performance of the network. As such, current orchestration frameworks are limited in their ability to provide optimal orchestration decisions across the different levels of the network.

3. **Limited AI/ML-Enabled Orchestration**: A third limitation of current orchestration frameworks proposed for 5G networks is their limited use of AI and ML technologies. Although such technologies are being proposed in current 5G network deployments [84,85], this is being done in a limited manner and only at specific levels within the network. This is partially due to the absence of real-world datasets that can be used to develop such AI/ML-enabled orchestration frameworks. As a result, these technologies are not being utilized to their full potential despite the fact that a large amount of important and relevant data can be collected by the different entities within the network. Consequently, this limits 5G networks' ability to truly become "zero-touch" networks (i.e., networks capable of self-configuration, self-monitoring, self-healing, and self-optimization [86]).

*3.3. Lessons Learned*

With the continued growth in the development and deployment efforts of 5G networks and infrastructures by various organizations and stakeholders, a list of lessons have been learned [87–90]. Whether it was related to the use cases considered, architectures deployed, or resources used, the information and results gathered represent a valuable resource to learn from when planning for 6G networks. Some of the lessons learned are as follows:

- Choice of 5G use cases and locations is critical [87]: Due to the fact that it is uncertain what type of 5G-compatible devices will be available and the type of services desired by customers, it is difficult to determine the type and location of 5G use cases and technologies to deploy. Hence, it is important to study and understand the customers' needs to be able to predict what 5G services are more likely to succeed and will result in higher revenue [87].

- Need for hybrid architecture for faster deployment [87]: To ensure faster deployments of 5G networks, hybrid architectures that can rely on 4G networks that have been built for greater capacity are needed [87]. This allows operators to build on their existing infrastructure and expanding their utilization, thus maximizing their benefit.

- Lucrativeness of enterprise use cases [87]: Studies have illustrated the lucrativeness of enterprise use cases, with almost 50% of operators participating in a recent poll having indicated that supporting new enterprise services is the main business driver for next-generation networks [87]. This does not come as a surprise given the market saturation for consumer mobile services. Accordingly, operators are focusing more on understanding the enterprise services' requirements to be able to cash in on this broad market [87].

- Key performance indicators for most organizations are speed, latency, and power efficiency [88]: Another recent poll conducted in Portugal indicated that the main performance metrics desired by organizations are speed, latency, and power efficiency. Accordingly, many of the companies believe 5G and WiFi 6 to be the most critical

wireless technologies within the next 3 years [88]. Therefore, these companies are focusing on understanding how to incorporate 5G technologies to provide the maximum financial return.

- Automation and simplification are pillars of success [89]: Recent new technology trials conducted by various operators and stakeholders have provided great insights. These trials showed that there is a need to automate activity and simplify the design to achieve greater throughput and more predictable cycle times [89]. In turn, this can help in unlocking and realizing 5G's full potential [89].
- Network resource and spectrum sharing is essential [90]: Due to the high cost associated with deploying 5G networks, policy makers and stakeholders have advocated for the sharing of network resources to minimize the build cost [90]. Similarly, they have advocated for the sharing of spectra between different operators to further reduce costs and encourage cooperation between network operators [90].
- Government assistance and support is crucial [90]: Assistance and support from governments is crucial to ensure that 5G and next-generation networks reach their full potential. Accordingly, it is extremely important for governments to provide funding for next-generation network research and development efforts. Moreover, governments should provide additional incentives for network operators to make sure that these technologies are incorporated into all industries in which they can contribute [90].

## 4. 6G End-to-End Network Automation Challenges

As shown earlier, 6G networks consist of complex infrastructure and a diverse set of heterogeneous services and fragmented use cases. Moreover, they have more stringent performance requirements. As a result, 6G end-to-end network automation and orchestration emerges as a prime concern to consider. Within the context 6G end-to-end network automation and orchestration, a fresh set of challenges exist that need to be investigated to ensure that these networks fulfill their potential. Figure 4 summarizes these challenges.

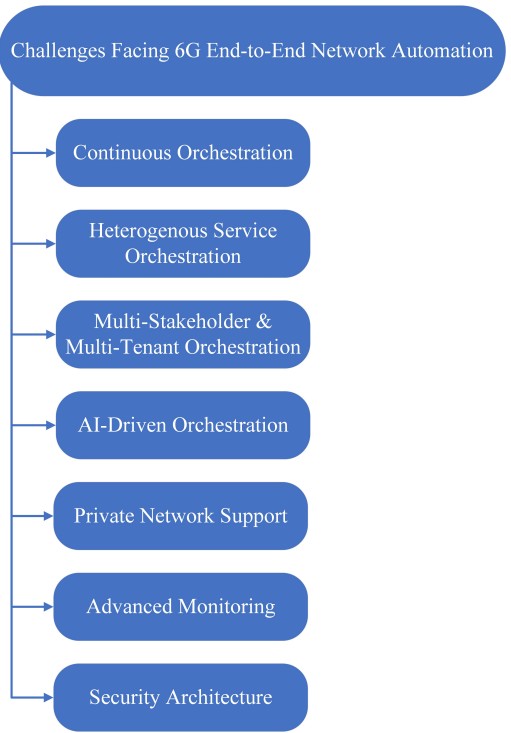

**Figure 4.** The 6G end-to-end network automation challenges.

### 4.1. Continuous Orchestration

The first challenge facing 6G network automation and orchestration is the need for continuous end-to-end (E2E) orchestration starting from the end devices (commonly referred to as extreme edge) to the edge nodes all the way to the cloud. This means that there needs to be continuous orchestration and resource management at three levels, namely the extreme edge, edge, and the cloud. The challenge lies in the fact that most current management and orchestration (MANO) frameworks do not extend to the extreme edge. More specifically, the current orchestration solutions rely on ETSI's NFV MANO framework [81,82]. This framework has been extended to handle edge nodes through dedicated/specialized virtual infrastructure managers (VIMs). However, these VIMs are still under the control of the centralized orchestrator. Therefore, current solutions are not yet suitable and applicable to the extreme edge.

In addition to the MANO framework itself, other factors need to be considered when developing the continuous E2E orchestration for 6G networks. The first is the heterogeneous nature of the extreme-edge devices that will be orchestrated. More specifically, 6G networks are expected to handle personal (e.g., smartphones, laptops, wearables, etc.), commercial (e.g., smart home appliances, smart speakers, etc.), and industrial (e.g., manufacturing robots, drones, etc.) devices [91]. This means that any MANO framework needs to be able to cater to the capabilities of these devices. Additionally, these devices will use a variety of supporting technologies and will have different operating systems, underlying hardware, and various interfaces. As such, the orchestration framework needs to be aware of the resources available for these supporting technologies. Lastly, the extreme-edge devices may be located in uncontrolled environments. Hence, the corresponding resources in such environments may not always be available. This is challenging, particularly in terms of ensuring orchestration continuity given the global extreme coverage requirement that 6G networks are expected to meet.

### 4.2. Heterogeneous Service Orchestration

The second challenge facing 6G network automation and orchestration is the heterogeneity of service definitions. This stems from the notion of the convergence of the physical, human, and digital worlds that 6G aims to achieve. As a result, new and heterogeneous services with varying performance targets emerge [92,93]. One aspect of these services is the multitude of underlying access technologies that will support them due to the multi-connectivity requirement in 6G networks. For example, 6G networks allow the combination and integration of fixed access with wireless access technologies. Therefore, 6G orchestration frameworks need to be able to optimally select the suitable interface in the integrated access network in a dynamic manner while accounting for each technology's infrastructure availability, capability, and topology. This is particularly important given the fact that current interface selection frameworks (through network slice templates and service descriptors) are static and infrastructure-agnostic. Therefore, it is desired that the instantiation of these new services is customizable by allowing the automatic adaptation to the network capabilities, resource availability, and the adopted technologies at the corresponding nodes. This is critical to ensure that the service deployment process is optimized and enhances the user's Quality of Experience (QoE) across multiple and possibly changing network infrastructures.

In addition to the above, 6G orchestration frameworks should also be able to support a richer and more versatile set of software-based components. This means that such frameworks need to be able to handle combinations of service definitions that use varying software modules, including virtual machines, containers, microservices, and serverless functions, simultaneously. Moreover, the orchestration frameworks need to handle different service decomposition patterns and functional splits. This is important in order to be able to define and offer the different required fine-grained services within the network continuum. Furthermore, the extreme-edge devices need to be considered as part of the 6G network slicing mechanisms to ensure that E2E orchestration is offered. This means that

the 6G orchestration frameworks should be able to aggregate the necessary resources (APIs, component definition primitives, service definition primitives, etc.) from the different stakeholders while still maintaining proper isolation between them.

### 4.3. Multi-Stakeholder and Multi-Tenant Orchestration

The third challenge facing 6G network automation and orchestration is the multi-stakeholder and multi-tenant orchestration. This stems from the fact that operators want to have a generic and open platform that combines all the assets and capabilities, including existing ones as well as new ones introduced by 6G networks. This is because large-scale service providers (such as Netflix and IBM) are adopting a "design-once-deploy-everywhere" approach [94]. This allows them to have their assets and capabilities consistently available across networks and technologies. As a result, new initiatives, such as the Open Platform Group (OPG) initiative launched by the GSMA alliance, have emerged to support this concept [95,96]. These lead to new services and value chains being created by the collaboration and cooperation of different stakeholders (e.g., communication service providers, over-the-top service providers, local and global cloud providers, private network operators, etc.). However, this brings a major concern in terms of orchestrating this cross-stakeholder collaboration. This is because the orchestration process has a direct impact on the monetization and federation opportunities of these stakeholders' capabilities. Moreover, 6G orchestration frameworks are expected to provide automated, secure, and trustworthy coordination and access to the stakeholders' capabilities and available resources. This includes resources available at the different portions of the network (i.e., cloud, edge, and extreme edge).

In addition to the multi-stakeholder notion, another related challenge is multi-tenant orchestration [97,98]. More specifically, ensuring that different network slices as well as different virtual network operators are logically separated is a challenging task. This is due to the fact that this separation between the different tenants and slices needs to be guaranteed. This can only be done through the continuous and effective monitoring and allocation of the available resources [98].

### 4.4. AI-Driven Orchestration

The fourth challenge facing 6G network automation and orchestration is how to leverage AI and ML technologies and for what tasks. This is particularly important given the large volume and heterogeneous nature of data that is expected to be generated by the core, edge, and specifically extreme edge devices. Although AI and ML technologies are being considered to manage and orchestrate current 5G network deployments [84,85], 6G requires that such technologies are used in a much more comprehensive manner that takes into account the architectural advancement of 6G networks.

One of the challenging aspects of incorporating AI/ML technologies into 6G network orchestration lies in determining what orchestration operations are being targeted. For example, AI/ML techniques can be used to predict required network scaling actions to ensure that traffic demands are met. Alternatively, these techniques can be used to take proactive network function placement decisions (whether at the core, edge, or extreme edge) based on the available resources as well as the requirements of the services formed by these functions. Moreover, they can be used to provide a proactive alerting system that interfaces with other systems and functions. Therefore, deciding on what orchestration operations need to be driven by these AI/ML technologies and where to place the corresponding module is a key issue to consider as part of the efforts of having AI-driven orchestration.

Another crucial and also challenging aspect is achieving data-driven cognitive management and orchestration—more specifically, the concept of closed-loop automation (CLA) that goes beyond current network analytics towards full cognitive and automated orchestration in the entire system. To achieve this, two main sub-challenges need to be considered. The first is the need for high-precision fine-grained data that can facilitate accurate decision-making. Thus, it is crucial to develop smart data collection mechanisms that only gather

the data needed for proper analytics and monitoring purposes. The second sub-challenge is facilitating cooperative and closed-loop mechanisms. This is particularly important given the fact that orchestration operations span across multiple domains and slices. Hence, any developed CLA mechanism needs to allow both the horizontal (across domains) and vertical (within a domain) interaction of closed loops as part of any data-driven approach.

A third essential challenging aspect that will also rely on data-driven approaches is intent-based orchestration. Intent can be defined as "the formal specification of all expectations including requirements, goals, and constraints given to a technical system" [99]. Accordingly, there is a differentiation between intent and the corresponding implementation logic of this intent. Therefore, to offer complete intent-based management and orchestration, multiple data-driven components that rely on AI and ML algorithms are needed. This includes components such as the intent interface, state modeling and anomaly detection, and self-learning closed loops, among others. Moreover, these components need to also facilitate cross-layer prediction and interaction across the different layers (e.g., infrastructure layer, management layer, data plane layer, etc.). This is important to ensure that these mechanisms are able to process and correlate the collected heterogeneous data.

Finally, an important aspect to consider when adopting and developing AI-driven orchestration mechanisms is explainability. Explainability is defined as making the AI systems'/agents' learning and decision-making process traceable and understandable for humans [100,101]. This is particularly important given that the decisions made by these learning agents have a direct impact on humans and, as such, humans should be able to trust and be convinced that these decisions are the correct ones. Within the context of service orchestration, it is important to develop mechanisms that humans can understand, particularly given the convergence between the digital, physical, and human worlds expected from 6G networks.

### *4.5. Private Network Support*

The fifth challenge facing 6G network automation and orchestration is how to support private networks in conjunction with the public networks. Currently, 5G supports private networks by deploying a 5G network slice completely on-premises or by offering it partially as a service [102,103]. In this case, network management is decoupled from deployment. As a result, provider agnostic APIs have been utilized to manage these private networks. However, 6G requires a deeper level of private network automation. More specifically, these management and orchestration APIs need to be able to better communicate and coordinate to enable better public–private network orchestration. To that end, there is a need for orchestration mechanisms that are able to effectively and flexibly combine internal enterprise capabilities with external functionalities to offer new services. Moreover, these mechanisms need to also be abstract to allow private network staff members to effectively manage the network without the need for deep knowledge in network communication, management, and orchestration.

### *4.6. Advanced Monitoring*

The sixth challenge that 6G network automation and orchestration mechanisms face is the need for continuous advanced monitoring. In this case, the monitoring process needs to collect metrics at different layers of the network architecture (i.e., at the core, edge, and extreme edge). This includes not only the infrastructure-related metrics (e.g., CPU consumption, RAM consumption, network usage, etc.) currently collected by 5G network monitoring modules, but also from the data plane in which the deployed service is running. For example, the capabilities/functionalities of virtual network functions such as the "Management Data Analytics Service (MDAS)" need to be augmented further and deployed at multiple layers. Similarly, functions such as the "Network Slice Selector Function (NSSF)" and the "Network function Repository Function (NRF)" need to be extended to contain "Slice Descriptor" and "Topology Descriptor" functions that provide additional information about the network state. This would allow for better network

slice elasticity decisions. Moreover, this can also contribute to enhanced real-time fault detection, alerting, and general diagnostics. Additionally, this would assist in describing the most optimized changes for blueprints in closed-loop automation. Accordingly, the 6G monitoring system needs to be able to:

1. Store real-time data generated by the different connected devices across the network continuum (at the core, edge, and extreme edge levels);
2. Offer real-time analysis and visualization of these data;
3. Take the corresponding right decisions in an autonomous and explainable manner;
4. Run in a decentralized and federated manner to support network management and service orchestration needs.

### 4.7. Security Architecture

Last but not least, the seventh challenge facing 6G network automation and orchestration is having a suitable security architecture. This is particularly important given the larger potential attack surfaces (due to the presence of the extreme edge devices) and the added stakeholders involved in 6G networks as compared to 5G networks [104,105]. One key parameter of the security of the 6G network and corresponding orchestration mechanisms is the Level of Trust (LoT) [106]. LoT has been identified as a key value indicator for 6G networks by the Hexa-X European project led by Ericsson and Nokia [107]. This indicator is important because 6G networks are expected to have a more pervasive impact on humans. Thus, humans need to trust that any system, technology, or infrastructure is secure and the corresponding decisions made by them are indeed the correct decisions.

Based on the aforementioned, any 6G orchestration framework or system needs to have several underlying security features. Firstly, the security mechanisms deployed need to be applicable system-wide and distributed across multiple autonomous managers. This helps to ensure that the response to any potential cyber-attacks is localized and performed in real time. Secondly, these mechanisms need to support a high level of collaboration and cooperation to ensure the E2E protection of all entities and stakeholders supported by the 6G orchestration framework. Thirdly, they need to be tailored to their deployment location and the placed load at the asset to be protected. Fourthly, the security mechanisms should also be capable of dealing with quantum computing-based threats given the rapid growth in this area. Finally, these mechanisms need to also be suitable for physical layer security and not just at the upper network and architecture layers.

In addition to these features, the impact of network slicing needs to be accounted for. This is crucial since any security compromise or anomaly in any of the resources or services can propagate to other dependent resources or services. In turn, this can compromise other network slices that may be sharing these resources and services. Accordingly, any 6G orchestration framework needs to take this into account, particularly if network slices are deployed across several domains and have differing administrative, technological, and organizational policies. Moreover, these security mechanisms need to be modular and autonomous in nature to ensure that they are scalable with the increasing number of network slices. Therefore, as illustrated above, the security aspect should not be neglected when designing and developing 6G orchestration frameworks since missing any security threat can have serious repercussions not only to the network and its entities, but also to the lives of the humans interacting with it.

## 5. Research Innovation Opportunities for Intelligent Automation

As illustrated above, the QoS requirements of 6G use cases are more strict and stringent. As a result, network service providers (NSPs) are required to meet these new requirements while also satisfying their business goals, thus creating often conflicting objectives. Accordingly, taking into account the increased complexity of these networks and aiming to achieve both these objectives, new innovation opportunities are created, particularly for network management and orchestration. More specifically, transitioning from a traditional network to an intelligent data-driven network supported by the abundance of data collected rep-

resents a promising viable solution for NSPs to meet the aforementioned objectives. To this end, ML is a prime candidate technology that can support NSPs by learning optimal patterns and policies from network-generated data and reduce the run-time complexity of traditional solutions [108,109]. In this section, some key research innovation opportunities for the intelligent orchestration of 6G networks are presented and summarized, as shown in Figure 5.

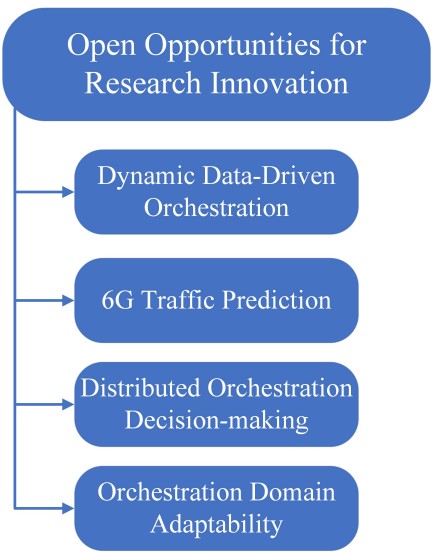

**Figure 5.** Research opportunities for innovation for intelligent orchestration.

### 5.1. Dynamic Data-Driven Orchestration

Although ML-based and ML-assisted network orchestration has been recently proposed in the literature, additional work can be done. This is because 6G networks are highly dynamic and thus are prone to constant change. Hence, there is a need for ML algorithms that can adapt to these changes. For example, ML algorithms need to consider the different QoS requirements and corresponding traffic priority levels that arise from network slicing and multi-tenancy, particularly with the continuous network service metric reporting, as well as the continuous resource slicing and monitoring. To that end, advanced intelligent paradigms and techniques such as reinforcement learning (RL) and federated learning (FL) can be viable solutions [110,111]. In particular, RL's ability to learn optimal policies through experience makes it a prime solution to be deployed as the basis of data-driven orchestration in 6G networks. This is because RL models can learn optimal orchestration policies and implement them while having a lower run-time complexity compared to other conventional ML techniques. In a similar fashion, FL's ability to develop a decentralized and privacy-preserving global model by relying on a highly distributed system makes it another prime potential solution towards the data-driven orchestration of 6G networks. Therefore, adopting such advanced paradigms and techniques will pave the way towards highly intelligent and automated 6G network orchestration.

Figure 6 is an illustrative example that shows an RL-enabled effective network orchestration framework. In this case, the RL agent collects data from the different terrestrial and non-terrestrial networks, particularly with respect to channel conditions and user density. Using this information, the RL agent can learn optimal resource allocation policies for the available networks. After learning these policies, the agent can then implement them for future decisions without being hindered by high computational complexity.

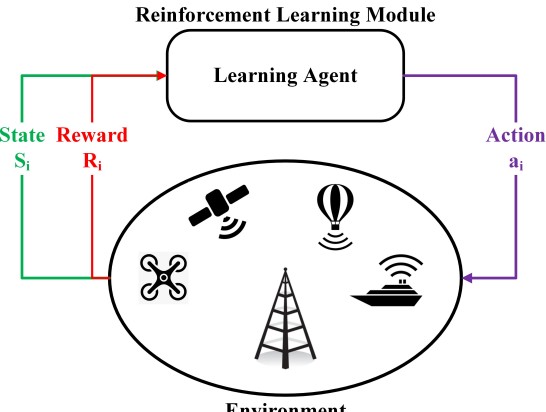

**Figure 6.** Potential RL-enabled 6G network orchestration architecture.

*5.2. 6G Traffic Prediction*

Another crucial component that has a direct impact on the orchestration process is the incoming traffic. Accordingly, being able to predict and forecast network traffic can be a game changer. This is where ML can play a significant role by accurately predicting the type, volume, and time of the incoming traffic. If the prediction is accurate, NSPs can ensure that the end-users' requirements are satisfied by planning their resource allocation and service provisioning decisions in advance. Moreover, network performance can be improved and the corresponding operation costs can be significantly reduced by optimizing the network traffic aggregation process, which can fully depend on the accurate prediction of the incoming traffic. Simultaneously, the impact of network slicing and multi-tenancy needs to be considered, particularly

- Ensuring traffic is separated between network slices and multi-tenant networks;
- Maintaining tenant authorization levels;
- Guaranteeing that the service level agreement requirements are met;
- Ensuring that data are isolated and authenticated.

In addition to the above, it is important to have a better understanding of both legacy networks and 6G network traffic's characteristics, particularly given the continuously changing and evolving end-user behavior. Accordingly, accurate traffic prediction models can be developed that can become key components of 6G network orchestration frameworks.

Figure 7 illustrates how ML can be incorporated at different levels within the network to enable accurate traffic prediction. For example, the ML-enabled traffic prediction module can be deployed near the network edge by allowing the different potential terrestrial and non-terrestrial 6G base stations (i.e., drones, satellites, roadside units, boats, etc.) to collect end-users'/devices' data, requesting different applications and services based on the network slice that they belong to. These base stations can then implement various supervised and semi-supervised ML models to predict the incoming traffic's type, volume, and time. Using this prediction, these base stations can then allocate the available wireless resources in a better and more effective manner. Moreover, these 6G network traffic predictions can be shared with the core network to better allocate the available resources at the transport level. In contrast, the ML-enabled traffic prediction module can be deployed at the 6G network core to reduce the computational complexity at the base stations (especially those with limited computing power, such as drones and roadside units). In this case, the ML-enabled traffic prediction module would collect the traffic generated by the different base stations to build the traffic prediction model. Then, this model can be used to better allocate the resources at the transport level, as well as assist base stations with their resource allocation decisions. The decision on where to deploy this traffic prediction module is determined by the NSPs' trade-off analysis between the computational complexity and the prediction granularity.

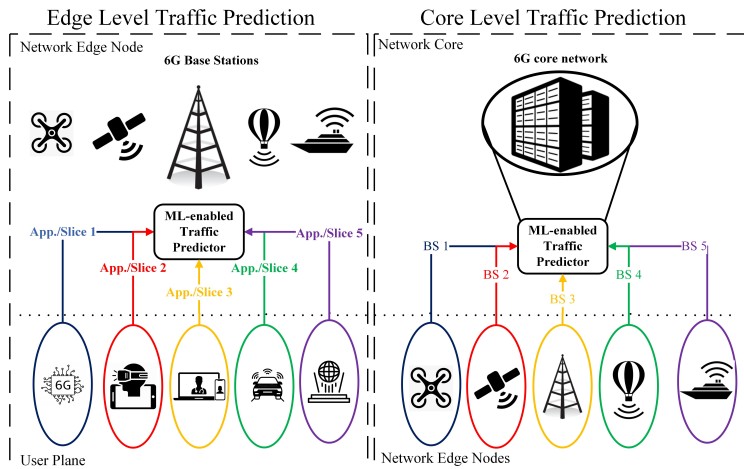

**Figure 7.** Potential ML-enabled 6G traffic prediction architectures.

### 5.3. Distributed Orchestration Decision-Making

Another concept that will contribute to effective and intelligent 6G network orchestration is distributed decision-making. This is particularly important given that the size and complexity of the network continues to grow. Therefore, there is a need to have a module capable of making rapid and intelligent decisions that leverage the data and insights gained from multiple regions and technologies. One viable architecture to adopt in this case is the FL architecture. This is because FL facilitates the cooperation between various intelligent agents without having them share the potentially sensitive data that they collected. In turn, these intelligent agents (which are placed at different locations throughout the network) can learn the characteristics of their underlying location or sub-network, as well as leverage the insights and intelligence from other locations and sub-networks. As a result, the performance, effectiveness, and efficiency of the network orchestration activities are increased.

Figure 8 illustrates a potential FL-enabled 6G orchestration architecture. In this case, local intelligent agents collect data about their region/sub-network and learn its corresponding model and parameters. These learned models and parameters are then shared with the FL global learning agent, which proceeds to aggregate them in order to develop the global model. As a result, better local network orchestration decisions can be made by leveraging the insights gained from all the regions and sub-networks constituting the 6G network.

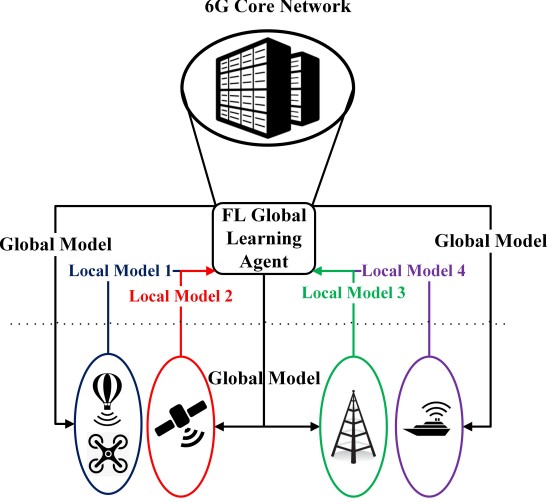

**Figure 8.** Potential FL-enabled 6G architecture.

*5.4. Orchestration Domain Adaptability*

The last opportunity related to intelligent 6G network orchestration is that of domain adaptability. This is particularly important given the highly dynamic and continually changing nature of 6G networks. As a result, it is expected that any ML-based model will not remain valid for all cases and scenarios. This phenomenon is known as model drift, in which the performance of the ML model degrades due to the change in the network status or environment. Accordingly, model drift detection mechanisms need to be incorporated into any ML implementation to alert NSPs that there is a need to remediate the ML model. This is particularly important given that new end-users, environments, sub-networks, and use cases continue to emerge, thus causing the underlying ML models to drift. Given that this concept has shown some promise in the network security case, incorporating it into network orchestration frameworks for 6G is essential to ensure the stability of such networks.

## 6. Conclusions

The digital transformation of businesses and services is now in full force, particularly given the current deployment efforts of 5G networks. This has opened the world to a new set of unique challenges and opportunities driven by the need for communication technologies and architectures that enable growth and sustainability while also promoting the digital transformation at hand. Here, 6G promises to be the set of technologies and architectures that achieves this goal by offering the means to interact and control the digital and virtual worlds that are decoupled from their physical location. Thus, 6G networks and architectures need to allow for the human experience to be expanded across the physical, biological, and digital worlds. To this end, a new set of enabling technologies (e.g., mmWave and THz communication, intelligent reflective surfaces, non-terrestrial and special purpose networks, and edge AI, among others) will be introduced for 6G in addition to those currently developed/deployed for 5G networks. Additionally, new concepts/paradigms that will emerge need to be considered, such as cell-less networks, tactile internet, and artificial intelligence (AI)-optimized wide-area networks.

As a result, a fresh set of challenges will emerge to enable the achievement of the desired 6G benefits. One such challenge is that of "end-to-end network automation". This is because such networks have to deal with more complex infrastructure and a diverse set of heterogeneous services and use cases. Therefore, orchestration processes need to be more proactive compared to the previously developed reactive methodologies adopted by previous network generations. Moreover, it is crucial to consider the role of AI and ML technologies as key enablers of predictive/proactive network slicing and orchestration.

To this end, this paper provided a vision on the role of different enabling technologies towards end-to-end intelligent automated 6G networks. Accordingly, this paper first reviewed the literature focusing on the orchestration and automation of next-generation networks by discussing in detail the challenges facing the efficient and effective management and orchestration of 6G Networks. This includes challenges such as continuous orchestration, heterogeneous service orchestration, and AI-driven orchestration, among others. Moreover, this paper presented some of the research opportunities for innovation within this area that aim at addressing the research gaps and tackling the aforementioned challenges. More specifically, it outlined how advanced data-driven paradigms such as reinforcement learning and federated learning can be incorporated into 6G networks for more dynamic, efficient, effective, and intelligent network automation and orchestration. This includes the use of RL for dynamic data-driven orchestration, ML for 6G traffic prediction at both the edge and core levels, and FL for distributed orchestration decision-making.

**Author Contributions:** Conceptualization, A.M.; Project administration, A.S.; Supervision, A.S. and A.A.-D.; Visualization, A.M.; Writing—original draft preparation, A.M.; Writing—review and editing, A.S. and A.A.-D. All authors have read and agreed to the published version of the manuscript.

**Funding:** This research received no external funding.

**Institutional Review Board Statement:** Not applicable.

**Informed Consent Statement:** Not applicable.

**Data Availability Statement:** Not applicable.

**Conflicts of Interest:** The authors declare no conflict of interest.

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
