# Peer review of "On End-to-End Intelligent Automation of 6G Networks"

_futureinternet, doi:10.3390/fi14060165_

Round 1

Reviewer 1 Report

The authors have put a lot of effort in this review writeup and is evident from the material that has been gathered from multiple sources. The intelligent automation is also very well written and provides a good overview of the possible uses in 6G.

However, the references are way too much than required and, at some occassions, authors have cited two references without a need. Published literature must be very carefully cited, and only for those occassions where the taken idea/text has no other way of justification or not too obvious. Also, more than 1 references is only required when the readers can't be satisfied with one reference (usually in a case where an unusual claim is made and two references are provided to advocate/support the text).

Secondly, review papers only benefit the readers when they cover all aspects of a topic. A good review of few technologies is much better than only touching several hundred of them and not being able to conclude or provide enough technical discussion. I'll only give one example from the paper and rest can be applied to other places in the paper:

page 7 (line 231 to 236) about THz: The information about its specturm is too obvious and availability of the large spectrum, and hence the possiblility of high data rates. But the review must mention some possible uses of it in 6G, use case scenarios and challenges in implementation and proposed solutions in the published literature. Just mentioning something without a proper review doesn't benefit anyone.
Then in the next paragraph, the authors talk about the high bandwidth available through OWC. Such information is already in abundant, and a comparision between THz technologies and OWC would have been useful. A lot of work has been done on the limitations of VLC based systems (such as LiFi). A review must had contained that information.

Author Response

Reviewer 1 Comments:

The authors have put a lot of effort in this review writeup and is evident from the material that has been gathered from multiple sources. The intelligent automation is also very well written and provides a good overview of the possible uses in 6G.

Thank you for valuable feedback.

  • However, the references are way too much than required and, at some occassions, authors have cited two references without a need. Published literature must be very carefully cited, and only for those occassions where the taken idea/text has no other way of justification or not too obvious. Also, more than 1 references is only required when the readers can't be satisfied with one reference (usually in a case where an unusual claim is made and two references are provided to advocate/support the text).

Thank you for your helpful comment. The reference list has been updated by removing ones where the ideas portrayed can be delivered with one reference. The removed references are highlighted via a red strikethrough.

  • Secondly, review papers only benefit the readers when they cover all aspects of a topic. A good review of few technologies is much better than only touching several hundred of them and not being able to conclude or provide enough technical discussion. I'll only give one example from the paper and rest can be applied to other places in the paper: page 7 (line 231 to 236) about THz: The information about its spectrum is too obvious and availability of the large spectrum, and hence the possibility of high data rates. But the review must mention some possible uses of it in 6G, use case scenarios and challenges in implementation and proposed solutions in the published literature. Just mentioning something without a proper review doesn't benefit anyone. Then in the next paragraph, the authors talk about the high bandwidth available through OWC. Such information is already in abundant, and a comparison between THz technologies and OWC would have been useful. A lot of work has been done on the limitations of VLC based systems (such as LiFi). A review must had contained that information.

Thank you for your insightful comment. This paper mainly focuses on reviewing the challenges facing end-to-end intelligent automation and orchestration of 6G networks that arise from the shortcomings of previous solutions as well as those introduced by the enabling technologies for 6G. Therefore, we do agree that providing a technical discussion about these technologies offers added benefit to the readers. Accordingly, we have enriched the discussion of many of the enabling technologies by presenting some of their possible use cases and implementation challenges. Moreover, we provided some comparison points between some of these technologies, particularly those that fall within the same category. Hence The following text has been added:

  1. Section 2.2.1. Wireless Communication Technologies (Pages 7 and 8):

• Millimeter Wave (mmWave) Communication: mmWave communication is defined as the communication using the frequency band ranging between 30 GHz to 300 GHz [32]. Due to the frequency band used, mmWave communication has the ability to offer Gbps speeds [33]. Accordingly, it has already been proposed as one of the technologies for 5G networks, particularly using frequencies below 50 GHz [35]. For example, mmWave has been proposed for potential deployment for enhanced local area access in 5G, achieving a peak data rate exceeding 10 Gbps [36]. Similarly, it has been proposed for cellular access for distances up to 200 m, achieving a 20x increase in capacity compared to 4G when utilizing directed antennas [36]. However, to support 6G and meet its performance requirements, further bands (such as above 100 GHz) will be needed to achieve the target throughput and latency values. More specifically, mmWave is expected to play a crucial role for short distance communication scenarios such as in autonomous/connected vehicles and smart factories communication [37].  It is worth noting that adopting mmWave introduces some challenges pertaining to the potential channel blocking by nearby entities (e.g. nearby vehicles in the case of autonomous/connected vehicles) which results in lower channel reliability.

  • Terahertz (THz) Communication: THz communication is defined as the communication using the frequency band from 0.1 - 10 THz [38]. Compared with mmWave, the benefit of using THz communication is two-fold. Firstly, there is an abundance of available spectrum in that frequency band (larger than that of mmWave communication), thus allowing for more users. Secondly, THz communication has the ability to support even higher throughput rates reaching Tbps [39]. This can be done without the need for any additional techniques for spectral efficiency enhancement [40]. Moreover, it offers improved system reliability, better energy efficiency, and higher robustness/adaptability to changing propagation scenarios [40]. Accordingly, THz communication is being proposed for multiple throughput-demanding delay-sensitive applications such as sensing, imaging, and localization [40]. Additionally, in a similar fashion to mmWave communication, THz communication is being proposed for a multitude of applications and use scenarios such as in local area networks, military communication applications, wireless data center networks, and space communication networks [41]. Despite the many benefits offered by THz communications, many corresponding challenges also arise. For example, one major challenge is the transceiver design and the high associated cost, particularly for the digital-to-analog and analog-to-digital converters (DACs and ADCs) [42]. Another challenge is the channel and noise modelling at the THz spectrum. This is because the channel characteristics in that frequency band significantly differs from other bands [42]. Similarly, due to the wavelength size, THz signals suffer from noise resulting from molecular absorption [42]. Hence, these concepts need to be considered when using THz communication for 6G.
  • Optical Wireless Communication (OWC)/Visible Light Communication (VLC): As the name suggests, optical wireless communication (OWC) is defined as the communication using the optical frequency band, namely between 300 GHz to 30 PHz [43]. This represents a rich source of bandwidth availability equivalent to 400 THz including the infrared, visible light, and ultraviolet sub-bands [43]. Among the potential OWC systems to be adopted, visible light communication (VLC) is considered to be an important supplementary technology to the current radio frequency communication systems. VLC is the communication using the frequency band between 400 THz to 800 THz [44]. Similar to the THz communication, the benefit of using VLC technology as a supplementary technology is two-fold. First, it offers high data transmission rates (up to 10 Gbps per wavelength [45]. Second, it can offer illumination solutions for indoor environments, particularly given the abundance of off-the-shelf optical devices available [45]. An added benefit of VLC that THz communication cannot offer is added security. More specifically, VLC can offer high physical layer security [46]. This is in contrast to THz communication which can only offer partial physical layer data security [47]. Due to its characteristics, VLC is being proposed as a viable solution in multiple applications such as indoor LiFi (light-based WiFi), vehicle-to-vehicle communication, and underwater communication to name a few [44]. Accordingly, OWC in general and VLC specifically promise to play a major role in 6G networks to enable and facilitate the achievement of the desired goals. This should be done by taking into account some of the limitations caused by the deployment of OWC and VLC-based systems such as frequent handovers, high inter-cell interference in dense deployments, high potential flickering, and the unsuitability for outdoor environments due to atmosphere-induced losses [48].
  1. Section 2.2.3 Network Architecture Technologies (Pages 9 and 10):

Software-Defined Networks (SDN): In addition to the aforementioned newly emerging networks, another enabling technology that already exists and will continue to play a pivotal role is Software-Defined Networks (SDNs). The basic premise of SDNS is separating the control and data planes using a centralized logical intelligence [68]. As a result, network management is facilitated, programmability is introduced, in addition to flexibility, scalability, and robustness being offered to the network [68]. Within the context of 6G, SDN continues to be adopted and proposed as a key network architecture choice [69]. More specifically, the SDN controller is being proposed as the “brain” within the control layer of 6G networks [69]. This component is being extended to the non-terrestrial portion of 6G networks such as space, air, and even sea-based networks. Hence, SDNs are still considered to be crucial to 6G networks by extending the paradigm to the new network types and components. However, a key challenge/issue to consider, particularly in 6G networks and the corresponding diverse sub-networks comprising it, is the choice of SDN deployment architecture. More specifically, choosing between a centralized controller architecture (better orchestration/management decisions at the expense of higher signaling overhead and less reliability) and a distributed controller architecture (less signaling overhead and better reliability at the expense of less optimal orchestration decisions) is a critical choice as it can have a direct impact on many of the desired performance metrics [71].

  1. Section 2.2.4 Other Integrated Technologies (Pages 10 and 11):

“Cloud native deployments: In addition to the aforementioned integrated technologies, cloud-native deployments are essential pillars for 6G networks. As per VMware’s definition “Cloud native is an approach to building and running applications that exploits the advantages of the cloud computing delivery model [78]”. This spans a multitude of technologies and tools including network function virtualization (NFV). NFV will continue to play a pivotal role in 6G networks [79][84]. Simply put, NFV can be defined as the migration process from dedicated hardware to software-based applications/containers/dockers that can run on standard commercial-of-the-shelf servers [85]. The benefit is that it improves network flexibility and scalability, reduces development cycles, lowers capital and operational expenditures (CAPEX and OPEX), and makes platforms more open [85].

Due to the characteristics of NFV paradigm and the corresponding benefits it offers, it has been proposed as a pillar and key enabler of a multitude of applications and services to be offered by 6G networks [86–88]. For example, various virtual network functions (VNFs) are being deployed as part of 6G network monitoring, resource management, and security modules of terrestrial and non-terrestrial 6G networks [86–88]. As such, NFV is and will continue to be an essential paradigm as part of any future 6G network deployment. This should be done while taking into account the different challenges introduced by adopting the NFV paradigm in 6G networks. This includes the functional and service-level considerations (placement and orchestration decisions), heterogeneity, and security considerations (redundancy and reliability) of virtual network functions [89].”

Reviewer 2 Report

The article presents a review on the development of 6G telecommunication systems from the point of view of solving the problems of orchestration of the resources of such networks, which are heterogeneous infocommunication systems. The management of such systems presents a significant technical challenge. The severity of this problem is shown by the authors in their review work.

As an overview, the article is certainly useful for getting an idea of the state of the art in the field of  automation in heterogeneous telecommunication systems.

Manuscript is clear, relevant for the field and presented in a well-structured manner.  The cited references are relevant. It does not include an excessive number of self-citations. The manuscript is scientifically sound.  The figures, tables and images are appropriate.

Author Response

Reviewer 2 Comments:

The article presents a review on the development of 6G telecommunication systems from the point of view of solving the problems of orchestration of the resources of such networks, which are heterogeneous infocommunication systems. The management of such systems presents a significant technical challenge. The severity of this problem is shown by the authors in their review work.

Thank you for your valuable feedback.

As an overview, the article is certainly useful for getting an idea of the state of the art in the field of  automation in heterogeneous telecommunication systems.

Thank you for your valuable feedback.

Manuscript is clear, relevant for the field and presented in a well-structured manner.  The cited references are relevant. It does not include an excessive number of self-citations. The manuscript is scientifically sound.  The figures, tables and images are appropriate.

Thank you for your valuable feedback.

Reviewer 3 Report

The abstract needs to be modified, the general part is unnecessarily long, and the specific purpose of the article is not clear from the text (it is described very generally). At the same time, the benefits and outputs are not clear from the abstract text to stimulate readers' interest to continue. I recommend shortening the general part and adding specific results.
I'm not convinced of the need for graphical representation in Figure 1. Readers certainly already know this. This is concise information.
Although the authors rely on the term Intelligent automation in the title and keywords, they do not work with this term. Likewise, it was appropriate to mention the IoT that the article concerns, at least in part. Here I would recommend, for example, authors in the field of automation, IoT, process digitisation, etc.:
https://doi.org/10.3390/s130100393
 https://doi.org/10.3390/s16101644
https://doi.org/10.3390/pr10030539
 https://doi.org/10.3390/su12198211
From the point of view of the management within which KPIs were created, they are misused in the article. I recommend deleting this term or replacing it with another term. KPIs are definable and measurable performance targets; this is not the case here. I have no comments on the selected areas, and they are appropriately named, but I would not call them KPIs.
A significant weakness of the paper is the absence of a methodology and a clearly defined goal. It is not clear from the text why the opinions used were used, based on which key they were selected and preferred over others. Within the Review of articles, the usual analysis of some databases, or a clear definition of the articles on which it is based, is all missing here.
At the end of the article, it is necessary to specifically mention the clear benefits of this article so that there is a clear contribution of the authors in summarising the views and ideas related to the issue of 6G.
The article provides a good overview of 6G networks but does not meet the usual requirements of a Scientific Review, especially in terms of methodology. The article rather popularises this term and brings this issue closer to the general public.

Author Response

Reviewer 3 Comments:

  1. The abstract needs to be modified, the general part is unnecessarily long, and the specific purpose of the article is not clear from the text (it is described very generally). At the same time, the benefits and outputs are not clear from the abstract text to stimulate readers' interest to continue. I recommend shortening the general part and adding specific results.

Thank you for your insightful comment. The abstract has been revised to reduce the general part and highlight the specific purpose of the paper which is to envision how end-to-end intelligent automated 6G networks can be achieved. This is done by first discussing the key enabling technologies of 6G networks along with some of the previously proposed solutions (particularly for 5G networks). It then delves deeper into the challenges facing 6G networks due to the limitations of previous works as well as the characteristics of the enabling technologies. Then, the paper outlines how advanced data-driven paradigms such as reinforcement learning and federated learning can be incorporated into 6G networks for more dynamic, efficient, effective, and intelligent network automation and orchestration.

Accordingly, the modified abstract is as follows (Page 1):

The digital transformation of businesses and services has been in full force, opening the world to a new set of unique challenges and opportunities. 6G promises to be the set of technologies, architectures, and paradigms that promote the digital transformation and enable growth and sustainability by offering them the means to interact and control the digital and virtual worlds that are decoupled from their physical location. One of the main challenges facing 6G networks is “End-to-End Network Automation”. This is because such networks have to deal with more complex infrastructure and a diverse set of heterogeneous services and fragmented use cases. Accordingly, this paper aims at envisioning the role of different enabling technologies towards end-to-end intelligent automated 6G networks. To that end, this paper first reviews the literature focusing on orchestration and automation of next-generation networks by discussing in detail the challenges facing efficient and fully automated 6G Networks. This includes automating both the operational and functional elements for 6G networks. Additionally, this paper defines some of the key technologies that will play a vital role in addressing the research gaps and tackling the aforementioned challenges. More specifically, it outlines how advanced data-driven paradigms such as reinforcement learning and federated learning can be incorporated into 6G networks for more dynamic, efficient, effective, and intelligent network automation and orchestration.

  1. I'm not convinced of the need for graphical representation in Figure 1. Readers certainly already know this. This is concise information.

Thank you for your comment. Figure 1 has been removed.

  1. Although the authors rely on the term Intelligent automation in the title and keywords, they do not work with this term. Likewise, it was appropriate to mention the IoT that the article concerns, at least in part. Here I would recommend, for example, authors in the field of automation, IoT, process digitisation, etc.:

https://doi.org/10.3390/s130100393
https://doi.org/10.3390/s16101644
https://doi.org/10.3390/pr10030539

https://doi.org/10.3390/su12198211

Thank you for your insightful comment. Indeed, IoT sensing devices and the resulting automation and digitization of processes and services are prime examples of the paradigms and technologies that 6G is expected to interact with and support. Hence, it is important to provide a brief discussion of them as part of the paper. To that end, examples of these use cases and concepts has been added as part of the introductory discussion of the motivation and need for 6G networks. More specifically, these technologies and concepts play a pivotal role in having smart cities and smart manufacturing process (Industry 4.0). This includes a variety of services and applications such as defense and public safety as well as predictive maintenance and customized/personalized digital services.

Accordingly, the following text has been added (Page 2):

“6G networks and architectures need to allow for the human experience to be expanded across the physical, biological, and digital worlds. This is due to the emergence of new devices equipped with more intuitive interfaces and new sensing technologies, particularly with the continuous deployment of Internet of Things (IoT) devices [7]. More specifically, these new IoT devices are key components in building smarter cities that aspire to be greener and more sustainable [ 9]. This is done by deploying these sensing devices for a multitude of applications and use cases such as water and electricity distribution, healthcare, and transportation [9]. Additionally, these devices can play a pivotal role in automating and improving defense and public safety processes [10]. For example, firefighters can make use of IoT devices installed in buildings to detect fires in a faster and more efficient manner [10]. Similarly, sensors deployed on various platforms such as satellites, radars, and unmanned aerial vehicles (UAVs) can provide situational awareness for military personnel [10]. Moreover, 6G networks are expected to support next-generation industrial operations environment (including Industry 4.0) while considering performance metrics such as positioning, sensing, ultra-reliability, energy efficiency, and extreme real-time [11]. This includes facilitating predictive maintenance of machines in smart factories as well as offering automated, customized, and personalized digital services in such environments [13][14].”

  1. From the point of view of the management within which KPIs were created, they are misused in the article. I recommend deleting this term or replacing it with another term. KPIs are definable and measurable performance targets; this is not the case here. I have no comments on the selected areas, and they are appropriately named, but I would not call them KPIs.

Thank you for your insightful comment. The KPIs are illustrated under each of the areas identified. For example, in the case of time-sensitive communications, the desired KPI target is having the end-to-end latency < 1 ms. Simiarly, for the case of extremely high data rate/capacity, the KPI targets are peak data rate >100 Gbps and a 100x capacity increase. However, to better highlight the areas, they are renamed as “performance characteristics” rather than KPIs. This is to clarify that these are desired characteristics (e.g. supporting extreme high data rate, extreme low energy and cost, etc.) along with glimpse of the corresponding KPIs for each of these characteristics. The figure now is as follows:

  1. A significant weakness of the paper is the absence of a methodology and a clearly defined goal. It is not clear from the text why the opinions used were used, based on which key they were selected and preferred over others. Within the Review of articles, the usual analysis of some databases, or a clear definition of the articles on which it is based, is all missing here.

Thank you for your valuable comment. The main aim of the paper is to provide a vision on how end-to-end intelligent automated 6G networks can be achieved. Hence, the goal was not to only review what has been previously done, but rather discuss the previous work and its limitations to better motivate the challenges/issues facing the end-to-end intelligent automation of 6G networks. We do agree that providing insights into how these related works was chosen can help readers better appreciate its content. To that end, we defined the methodology adopted for identifying the works chosen, particularly those discussed in the related works portion of Section 3. More specifically, we searched the Scopus database to ensure that non-refereed publications are excluded. Moreover, only conference and journal papers between the years 2018 to 2022 from top-tier publishers such as IEEE, Springer, Elsevier, and MDPI were selected to highlight the state-of-the-art (SOTA) works in the area of 5G and network orchestration.

Accordingly, the following text was added (Page 11):

“As part of the related work exploration process, the Scopus database was used to ensure that non-refereed publications are excluded. As a result, the papers chosen are mainly from top-tier conferences and journals of reputable publishers such as IEEE, Springer, Elsevier, and MDPI. The search terms used are ``5G'' AND ``Orchestration'' with the search period limited to the years between 2018 and 2022 to represent the state-of-the-art (SOTA) works.”

  1. At the end of the article, it is necessary to specifically mention the clear benefits of this article so that there is a clear contribution of the authors in summarising the views and ideas related to the issue of 6G.

Thank you for your helpful comment. As mentioned earlier, this paper aims at envisioning how end-to-end intelligent automated 6G networks can be achieved. This is done by first discussing the key enabling technologies of 6G networks along with some of the previously proposed solutions (particularly for 5G networks). It then delves deeper into the challenges facing 6G networks due to the limitations of previous works as well as the characteristics of the enabling technologies presented. Then, the paper outlines how advanced data-driven paradigms such as reinforcement learning and federated learning can be incorporated into 6G networks for more dynamic, efficient, effective, and intelligent network automation and orchestration. More specifically, it describes how we can use RL for dynamic data-driven orchestration, ML for 6G traffic prediction at both the edge and core levels, and FL for distributed orchestration decision-making.

Accordingly, the last paragraph in the conclusion has been revised to summarize the contributions of the paper by outlining the views and ideas related to the issues in 6G as follows (Page 23):

“To that end, this paper provided a vision on the role of different enabling technologies towards end-to-end intelligent automated 6G networks. Accordingly, this paper first reviewed the literature focusing on orchestration and automation of next-generation networks by discussing in detail the challenges facing the efficient and effective management and orchestration of 6G Networks. This includes challenges such as continuous orchestration, heterogeneous service orchestration, and AI-driven orchestration among others. Moreover, this paper presented some of the research opportunities for innovation within this area that aim at addressing the research gaps and tackling the aforementioned challenges. More specifically, it outlined how advanced data-driven paradigms such as reinforcement learning and federated learning can be incorporated into 6G networks for more dynamic, efficient, effective, and intelligent network automation and orchestration. This includes the use of RL for dynamic data-driven orchestration, ML for 6G traffic prediction at both the edge and core levels, and FL for distributed orchestration decision-making.”

  1. The article provides a good overview of 6G networks but does not meet the usual requirements of a Scientific Review, especially in terms of methodology. The article rather popularises this term and brings this issue closer to the general public.

Thank you for your helpful comment. As mentioned earlier, the main goal of the paper is to provide a vision on how end-to-end intelligent automated 6G networks can be achieved. This is done by first discussing the key enabling technologies of 6G networks along with some of the previously proposed solutions (particularly for 5G networks). It then delves deeper into the challenges facing 6G networks due to the limitations of previous works as well as the characteristics of the enabling technologies presented. Then, the paper outlines how advanced data-driven paradigms such as reinforcement learning and federated learning can be incorporated into 6G networks for more dynamic, efficient, effective, and intelligent network automation and orchestration. Accordingly, the most suitable manuscript type from MDPI’s list based on the provided description was “Review” and hence it was used in this paper. In order to provide further insights/benefits to the reader and give it a better sense of a “Scientific Review”, the following has been done:

  1. We defined the methodology adopted for identifying the works chosen, particularly those discussed in the related works portion of Section 3. The following text was added (Page 11):

“As part of the related work exploration process, the Scopus database was used to ensure that non-refereed publications are excluded. As a result, the papers chosen are mainly from top-tier conferences and journals of reputable publishers such as IEEE, Springer, Elsevier, and MDPI. The search terms used are ``5G'' AND ``Orchestration'' with the search period limited to the years between 2018 and 2022 to represent the state-of-the-art (SOTA) works.”

  1. We provided a technical discussion about the enabling technologies by presenting some of their possible use cases and implementation challenges. Moreover, we provided some comparison points between some of these technologies, particularly those that fall within the same category. Hence The following text has been added:
    1. Section 2.2.1. Wireless Communication Technologies (Pages 7 and 8):

• Millimeter Wave (mmWave) Communication: mmWave communication is defined as the communication using the frequency band ranging between 30 GHz to 300 GHz [32]. Due to the frequency band used, mmWave communication has the ability to offer Gbps speeds [33]. Accordingly, it has already been proposed as one of the technologies for 5G networks, particularly using frequencies below 50 GHz [35]. For example, mmWave has been proposed for potential deployment for enhanced local area access in 5G, achieving a peak data rate exceeding 10 Gbps [36]. Similarly, it has been proposed for cellular access for distances up to 200 m, achieving a 20x increase in capacity compared to 4G when utilizing directed antennas [36]. However, to support 6G and meet its performance requirements, further bands (such as above 100 GHz) will be needed to achieve the target throughput and latency values. More specifically, mmWave is expected to play a crucial role for short distance communication scenarios such as in autonomous/connected vehicles and smart factories communication [37].  It is worth noting that adopting mmWave introduces some challenges pertaining to the potential channel blocking by nearby entities (e.g. nearby vehicles in the case of autonomous/connected vehicles) which results in lower channel reliability.

  • Terahertz (THz) Communication: THz communication is defined as the communication using the frequency band from 0.1 - 10 THz [38]. Compared with mmWave, the benefit of using THz communication is two-fold. Firstly, there is an abundance of available spectrum in that frequency band (larger than that of mmWave communication), thus allowing for more users. Secondly, THz communication has the ability to support even higher throughput rates reaching Tbps [39]. This can be done without the need for any additional techniques for spectral efficiency enhancement [40]. Moreover, it offers improved system reliability, better energy efficiency, and higher robustness/adaptability to changing propagation scenarios [40]. Accordingly, THz communication is being proposed for multiple throughput-demanding delay-sensitive applications such as sensing, imaging, and localization [40]. Additionally, in a similar fashion to mmWave communication, THz communication is being proposed for a multitude of applications and use scenarios such as in local area networks, military communication applications, wireless data center networks, and space communication networks [41]. Despite the many benefits offered by THz communications, many corresponding challenges also arise. For example, one major challenge is the transceiver design and the high associated cost, particularly for the digital-to-analog and analog-to-digital converters (DACs and ADCs) [42]. Another challenge is the channel and noise modelling at the THz spectrum. This is because the channel characteristics in that frequency band significantly differs from other bands [42]. Similarly, due to the wavelength size, THz signals suffer from noise resulting from molecular absorption [42]. Hence, these concepts need to be considered when using THz communication for 6G.
  • Optical Wireless Communication (OWC)/Visible Light Communication (VLC): As the name suggests, optical wireless communication (OWC) is defined as the communication using the optical frequency band, namely between 300 GHz to 30 PHz [43]. This represents a rich source of bandwidth availability equivalent to 400 THz including the infrared, visible light, and ultraviolet sub-bands [43]. Among the potential OWC systems to be adopted, visible light communication (VLC) is considered to be an important supplementary technology to the current radio frequency communication systems. VLC is the communication using the frequency band between 400 THz to 800 THz [44]. Similar to the THz communication, the benefit of using VLC technology as a supplementary technology is two-fold. First, it offers high data transmission rates (up to 10 Gbps per wavelength [45]. Second, it can offer illumination solutions for indoor environments, particularly given the abundance of off-the-shelf optical devices available [45]. An added benefit of VLC that THz communication cannot offer is added security. More specifically, VLC can offer high physical layer security [46]. This is in contrast to THz communication which can only offer partial physical layer data security [47]. Due to its characteristics, VLC is being proposed as a viable solution in multiple applications such as indoor LiFi (light-based WiFi), vehicle-to-vehicle communication, and underwater communication to name a few [44]. Accordingly, OWC in general and VLC specifically promise to play a major role in 6G networks to enable and facilitate the achievement of the desired goals. This should be done by taking into account some of the limitations caused by the deployment of OWC and VLC-based systems such as frequent handovers, high inter-cell interference in dense deployments, high potential flickering, and the unsuitability for outdoor environments due to atmosphere-induced losses [48].
  1. Section 2.2.3 Network Architecture Technologies (Pages 9 and 10):

Software-Defined Networks (SDN): In addition to the aforementioned newly emerging networks, another enabling technology that already exists and will continue to play a pivotal role is Software-Defined Networks (SDNs). The basic premise of SDNS is separating the control and data planes using a centralized logical intelligence [68]. As a result, network management is facilitated, programmability is introduced, in addition to flexibility, scalability, and robustness being offered to the network [68]. Within the context of 6G, SDN continues to be adopted and proposed as a key network architecture choice [69]. More specifically, the SDN controller is being proposed as the “brain” within the control layer of 6G networks [69]. This component is being extended to the non-terrestrial portion of 6G networks such as space, air, and even sea-based networks. Hence, SDNs are still considered to be crucial to 6G networks by extending the paradigm to the new network types and components. However, a key challenge/issue to consider, particularly in 6G networks and the corresponding diverse sub-networks comprising it, is the choice of SDN deployment architecture. More specifically, choosing between a centralized controller architecture (better orchestration/management decisions at the expense of higher signaling overhead and less reliability) and a distributed controller architecture (less signaling overhead and better reliability at the expense of less optimal orchestration decisions) is a critical choice as it can have a direct impact on many of the desired performance metrics [71].

  1. Section 2.2.4 Other Integrated Technologies (Pages 10 and 11):

“Cloud native deployments: In addition to the aforementioned integrated technologies, cloud-native deployments are essential pillars for 6G networks. As per VMware’s definition “Cloud native is an approach to building and running applications that exploits the advantages of the cloud computing delivery model [78]”. This spans a multitude of technologies and tools including network function virtualization (NFV). NFV will continue to play a pivotal role in 6G networks [79][84]. Simply put, NFV can be defined as the migration process from dedicated hardware to software-based applications/containers/dockers that can run on standard commercial-of-the-shelf servers [85]. The benefit is that it improves network flexibility and scalability, reduces development cycles, lowers capital and operational expenditures (CAPEX and OPEX), and makes platforms more open [85].

Due to the characteristics of NFV paradigm and the corresponding benefits it offers, it has been proposed as a pillar and key enabler of a multitude of applications and services to be offered by 6G networks [86–88]. For example, various virtual network functions (VNFs) are being deployed as part of 6G network monitoring, resource management, and security modules of terrestrial and non-terrestrial 6G networks [86–88]. As such, NFV is and will continue to be an essential paradigm as part of any future 6G network deployment. This should be done while taking into account the different challenges introduced by adopting the NFV paradigm in 6G networks. This includes the functional and service-level considerations (placement and orchestration decisions), heterogeneity, and security considerations (redundancy and reliability) of virtual network functions [89].”

Round 2

Reviewer 3 Report

The authors have very appropriately incorporated all comments and, in this way, have significantly improved the submitted text. I have no further comments.